



# Kinematics and time-resolved evolution of the main thrust-sense shear zone in the Eo-alpine orogenic wedge (the Vinschgau Shear Zone, Eastern Alps)

5    Chiara Montemagni[1], Stefano Zanchetta[1], Martina Rocca[1], Igor M. Villa[1], Corrado Morelli[2], Volkmar Mair[2], Andrea Zanchi[1]

1 Dipartimento di Scienze dell'Ambiente e della Terra, Università degli Studi di Milano – Bicocca, Milano, 20126, Italia
Ufficio Geologia e Prove Materiali, Provincia Autonoma di Bolzano Alto Adige, Cardano, 39053, Italia
*Correspondence to*: chiara.montemagni@unimib.it

**Abstract**

The Vinschgau Shear Zone (VSZ) is one of the largest and most significant shear zones developed in plastic conditions within the Austroalpine domain, juxtaposing the Ötztal and the Texel units to the Campo, Scharl and Sesvenna units

during the building of the Eo-alpine Orogen. The VSZ dominates the structural setting of a large portion of the central Austroalpine Late Cretaceous thrust stack. In order to fully assess the evolution of the VSZ, a multi-faceted approach based on detailed multiscale structural and petrochronological analyses has been carried out across representative transects of the shear zone in the Vinschgau Valley. The research has been performed with a view to characterizing kinematics, *PT* conditions and age of motion of the VSZ.

Our fieldwork-based analyses suggest that the dip of mylonitic foliation increases from W to E with an E-W trending stretching lineation which dips alternatively to the W and to the E, due to later folding related to the Cenozoic crustal shortening. The dominant top to the W-directed shear sense of the mylonites recognized in the field and confirmed by microstructural analyses is in agreement with the exhumation of the upper Austroalpine nappes in the hanging wall of the shear zone: the Texel unit with Late Cretaceous eclogites, the Schneeberg and Ötztal units both affected by Eo-alpine

amphibolite-facies metamorphism. Chemical and microstructural analyses suggest deformation temperatures of ca. 350-400 °C during shearing. Timing of deformation along the VSZ has been constrained for the first time through $^{40}Ar/^{39}Ar$ dating of syn-shearing micas, which reveal a Late Cretaceous age of the VSZ mylonites with ages ranging between 80 and 97 Ma. A systematic younging trend of deformation occurs towards the central part of the shear zone in the studied transects. Vorticity analysis shows a clear decrease in the simple shear component correlated to the younging direction of

mica ages towards the core of the shear zone. This evolution is consistent with the growth of a shear zone where strain localizes into its central part during deformation. The defined evolution of the VSZ sheds new light on how large-scale thrust-sense shear zones act and how much exhumation they can accommodate in the frame of an evolving orogenic wedge.

## 1.  Introduction

The Vinschgau Shear Zone (VSZ), extending along the homonymous valley (NE Italy), is one of the most important tectonic structures developed within the Austroalpine domain of the Alps (Fig. 1 and Fig. 2). Starting from the first systematic studies of the Alpine belt, this large shear zone, firstly defined as the "Schlinig Thrust", was interpreted as a top-to-W thrust plane (Spitz and Dyhrenfurth, 1914). Although several authors later rejected this interpretation (Heim,

1922; Staub, 1937), modern studies (Schmid and Haas, 1989; Froitzheim et al., 1994, 1997) carried out since the end of the last century demonstrated the validity of the first assumptions. Recent structural analyses (Brunel, 1980; Ratschbacher, 1986, Ratschbacher et al., 1991; Schmid and Haas, 1989; Pomella et al., 2016) recognized indeed that the entire central Austroalpine nappe stack was affected by W-directed tectonic transport during the first stages of the Late Cretaceous Alpine deformations. Along the VSZ, the Austroalpine tectonometamorphic units with a dominant metamorphism of



Alpine age overthrust Austroalpine units that were only slightly affected by Alpine metamorphism (up to greenschist facies) and deformation, still largely preserving features acquired during the Variscan orogeny.

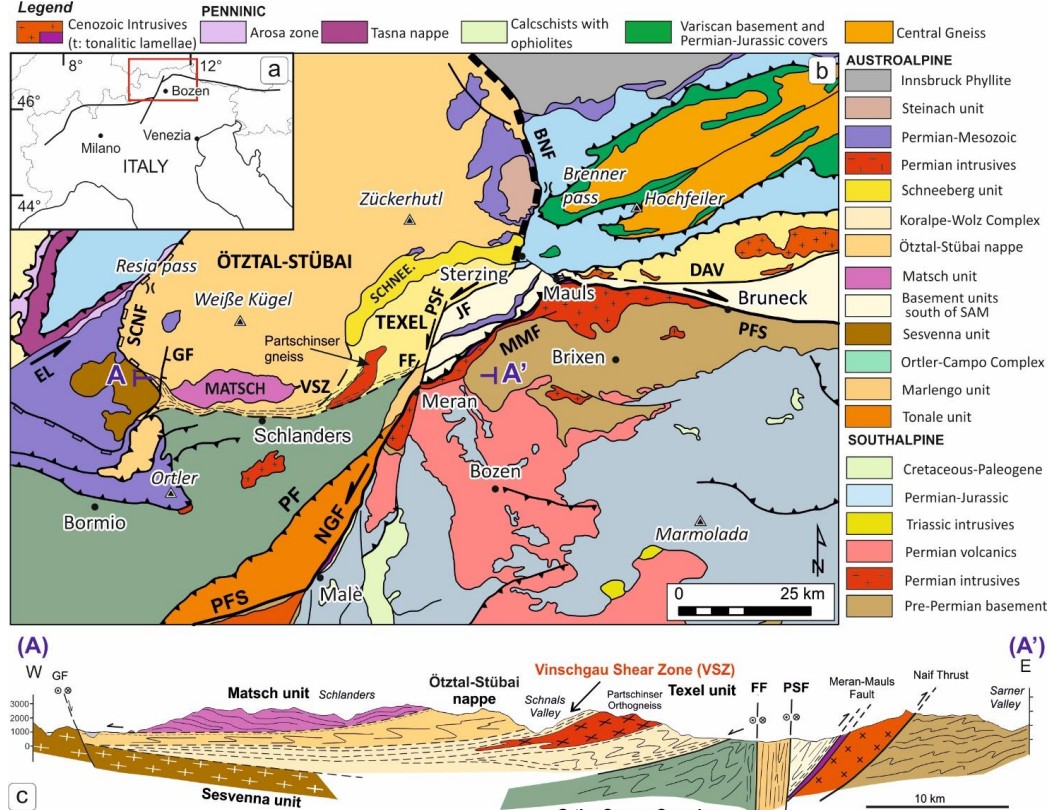

**Figure 1: Geological setting of the eastern-central Alps. (a) Northern Italy, location of the study area is marked with a red box; (b) Tectonic scheme of the eastern-central Alps (modified after Schmid et al., 2004; Pomella et al., 2016). (c) The cross section,**
**trace A-A' in figure (a), has been drawn in a larger scale with respect of the tectonic scheme. BNF: Brenner Normal Fault; DAV: Defereggen-Anthoslz-Vals Fault; EL: Engadine Line; FF: Forst Fault; GF: Glurns Fault; JF: Jaufen Fault; NGF: North Giudicarie Fault; PF: Pejo Fault; PFS: Periadriatic Fault; PSF: Passeier Fault; SCNF: Schlinig Normal Fault; VSZ: Vinschgau Shear Zone.**


The VSZ is almost continuously exposed for more than 50 km, mainly on the northern flank of the Vinschgau Valley (Fig. 1), reaching a maximum thickness of about 550 m close to Eyrs (Fig. 1 and Fig. 2). These features make the VSZ one of the largest ductile thrust-sense shear zone now exposed in the Alps, together with the Periadriatic Fault (Schmid et al., 1987; 1989) and the Orobic Thrust in the Southern Alps (Zanchetta et al., 2011; D'Adda and Zanchetta, 2015). Due
to its complete exposure and accessibility, this prominent shear zone represents an ideal natural laboratory for the study of finite strain distribution during the development of a large mature intra-basement shear zone and to evaluate the evolution in terms of finite strain localization, coaxiality, kinematic, and lifetime of activity (e.g. Xypolias, 2010; Xypolias and Koukouvelas, 2001; Law et al., 2013; Fossen and Cavalcante, 2017; Oriolo et al., 2016, 2018).

Large scale thrust- or normal-sense shear zone developed within collisional setting display huge along-strike exposures
as the ca. 2500 km of the Main Central Thrust and South Tibetan Detachment in the Himalayan orogen (e.g. Caby et al., 1983; Searle et al., 2008), the ca. 30 km of the Simplon Shear Zone (Mancktelow, 1985) and Brenner Fault (Rosenberg



et al., 2018) in the Alps or the Great Slave Lake shear zone (Hamner et al., 1992) in northern Canada. Due to its peculiar exposure along strike, the VSZ shows different features from shallow depths conditions at its western portion and deeper conditions at the eastern end (Schmid and Haas, 1989), providing a complete crustal section of a large scale shear zone.

This kind of exposure provides insights not only on the different deformation mechanisms and behaviours of the shear zone at different crustal levels, but also on its evolution through time.

In this work, we applied a quantitative approach to reconstruct the evolution through space (depth) and time of the VSZ. $^{40}Ar/^{39}Ar$ dating of syn-mylonitic micas sampled along several transects of the VSZ have been combined with microstructural and mineralogical analyses, aimed to define the *PT* condition of shearing. The analysis of the finite strain

distribution and the kinematic vorticity of flow (e.g. Montemagni and Zanchetta, 2022; Petroccia et al., 2022) finally provide a full kinematic and time-resolved evolution of the VSZ, that could be taken as an example of how a large-scale thrust-sense shear zone develop within a collisional orogen.

## 2.    Geological setting

The study area (Fig. 1 and Fig. 2) is located along the Vinschgau Valley (NE Italy) entirely extending within the central Austroalpine domain, comprised between the Northern Calcareous Alps to the north and the Periadriatic Fault to the south (Fig. 1). Here, the Austroalpine domain consists of tectono-metamorphic units that have been identified based on paragenesis, deformation history, metamorphism and relative ages. These units are the Pejo and Laas units (belonging to the Campo-Ortler nappe system), the Ötztal-Stubai Complex, the Matsch unit, and, finally, the Texel and the Schneeberg

units, belonging to the Koralpe-Wölz high-pressure nappe system (Schmid et al., 2004; Handy et al., 2010; Pomella et al., 2016). The E-W striking VSZ separates the Pejo and Laas units to the south, the footwall of the VSZ characterized by greenschist facies Alpine metamorphism, from the Ötztal, Matsch, Texel and Schneeberg units forming the hanging wall, characterized by amphibolite to eclogite facies Alpine metamorphism (Fig. 2). Therefore, the VSZ together with the Passeier Fault, the Jaufen Fault and the Defereggen-Antholz-Vals Fault (PSF, JF and DAV in Fig. 1) has been considered

to form the Southern limit of Alpine Metamorphism (SAM, Hoinkes et al., 1999), a large faults system defining the southern border of the high-grade Alpine metamorphism in the Austroalpine domain of the Eastern Alps.

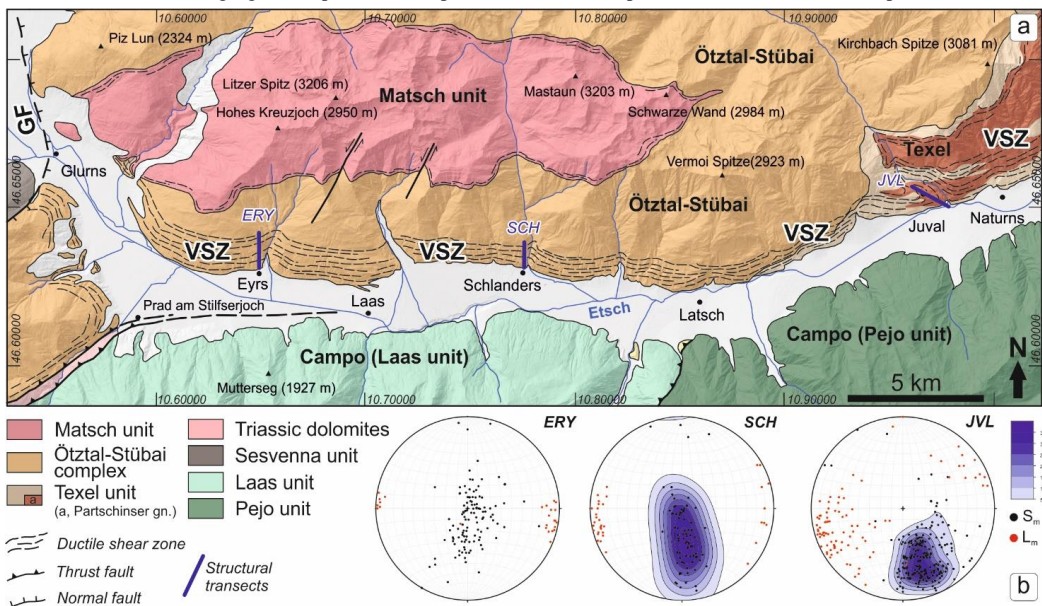

**Figure 2: (a) Tectonic scheme of the Vinschgau Valley, the location of studied transects (ERY, SCH and JVL) has been reported. (b) Structural data of Eyrs, Schlanders and Juval transects, equal area, lower hemisphere stereographic projections of**

**mylonitic foliation ($S_m$) and lineation ($L_m$). GF: Glurns Fault; VSZ: Vinschgau Shear Zone.**



The VSZ has been described as a ductile-to-brittle fault formed by a thick zone of mylonites and phyllonites exposed mainly along the left hydrographic side of the Vinschgau Valley (Fig. 2; Schmid and Haas, 1989; Conti, 1994; Froitzheim et al., 1997; Thöni, 1999; Sölva et al., 2005; Pomella et al., 2016; Koltai et al., 2018; Klug and Froitzheim, 2022). Schmid

and Haas (1989) defined the main structure as a thick intrabasement shear zone dominated by intracrystalline plastic processes, showing different thermal conditions ranging from 300 °C to the west up to 500 °C to the east. The VSZ forms a wide system of shear zones, branching out eastward across the folded boundaries between the Ordovician Partschinser Granodiorite (also named Tschigot orthogneiss, Zantedeschi, 1991) and the host paragneiss of the Texel unit. The VSZ shows a gently N-dipping foliation (20°-30°) and a constantly E-W trending lineation, with kinematic indicators at the

mesoscopic and microstructural scale that point to a top-to-N/NW sense of shear (Schmid and Haas, 1989).

Four tectonic units mainly consisting of polyphase metamorphic crystalline basement rocks, which have been deeply involved in the Alpine deformation and metamorphism, form the hanging wall of the shear zone. They consist of the western termination of the Texel and Schneeberg units, and of the Ötztal unit, one of the largest nappes of the Late Cretaceous Austroalpine thrust stack, which is overthrust by the Matsch unit, forming a folded klippe atop of the VSZ

mylonites.

The age pattern of the Alpine metamorphic peak of the tectonometamorphic units in the hanging wall of the VSZ displays almost coeval ages in the Texel and the Schneeberg units. Partially amphibolitized eclogite boudins, preserved within the micaschists and paragneisses of the Texel unit, point to metamorphic peak conditions during the Alpine orogenesis of 540-630 °C and 1.2-1.4 GPa (Habler et al., 2006), with even higher temperature and pressure suggested for other eclogite

occurrences (Poli, 1991; Zanchetta et al., 2012; Zanchetta et al., 2013). Peak metamorphic conditions close to the amphibolite/eclogite facies boundary have been suggested for the Schneeberg unit, with peak temperature of 550-600 °C, at pressure of 0.8-1.0 GPa (Konzett and Hoinkes, 1996; Krenn et al., 2011). Geochronological data point to a Late Cretaceous age for both the Texel eclogites (U-Pb zircon ages of $84 \pm 5$ Ma; Habler et al., 2006; Zanchetta et al., 2013) and the upper amphibolite facies metamorphism of the Scheneeberg unit, for which ages ranging between 85-86 Ma

($^{40}Ar/^{39}Ar$ age on paragonitic white mica in amphibolites, Konzett and Hoinkes, 1996) and $90.9 \pm 4.1$ Ma (Sm-Nd on garnet cores in metapelites, Sölva et al., 2005) have been obtained.

The age and the peak metamorphic conditions experienced by the Ötztal unit during the Alpine metamorphism are far less constrained. Middle to Late Cretaceous K-Ar mica ages (100-110 Ma) have been obtained by Thöni (1980) from the basement in the hanging wall of the VSZ and a whole rock Rb-Sr age of $83 \pm 1$ Ma resulted from a metapegmatite of the

Matsch unit (Fig. 2) at the western end of the VSZ (Thöni, 1986). An Alpine age of metamorphism of the Matsch unit has been also argued by Habler et al. (2009), on the base of Early to Middle Permian intrusions age of pegmatites, later deformed and metamorphosed. The Alpine metamorphism in the Ötztal reached 530-550 °C in the southeastern part (Purtscheller and Rammlmair, 1982; Hoinkes et al. 1999), with temperatures that progressively decrease toward NW. Pressure estimates are substantially lacking. On the base of newly formed mineralogical assemblages in pre-Alpine

andesitic and andesitic/basaltic dikes intruding the Ötztal basement, Purtscheller and Rammlmair (1982) supposed a maximum pressure of about 0.5-0.6 GPa, confirmed by recent data (Zanchetta et al., 2013). Additional *PT* estimates have been proposed by Gregnanin and Valle (1995) for the Ötztal metasedimentary cover (550 °C and 1.0 GPa), and by Tropper and Reichis (2009) for the SW part of the Ötztal basement (560 °C and 0.88 GPa) close to the Schneeberg unit.

The westernmost portion of the VSZ is crosscut by the Glurns Fault (GF in Fig. 2), along which the Ötztal basement is in

contact with the Sesvenna unit, mainly consisting of orthogneiss with a Variscan medium-grade metamorphic imprint and its Permian-Mesozoic sedimentary cover (Froitzheim et al., 1994; 1997).

### 3. Structural analysis
#### 3.1. General description and methods

The entire VSZ has been individuated and followed in the field from Naturns (E) to Glurns (W) (Fig. 2). The maximum thickness is reached close to its western ends, at Eyrs, where it is estimated to be of about 600 m. To the east of Naturns (Fig. 2) the VSZ widens and branches out in several shear zones that wrap around the rigid body of the Partschinser orthogneiss of the Texel unit, as already noticed by Schmid and Haas (1989), which has given a Rb-Sr radiometric age of





about 450 Ma (Zantedeschi, 1991). Here, in the Meran area, the VSZ is crosscut by most recent shear zones and faults
(Bargossi et al., 2010; Zanchetta et al., under review).

Detailed field structural analyses and sampling of the VSZ were performed along three selected transects (Fig. 2) in the localities, from E to W: Juval, Schlanders and Eyrs. The three studied geological sections were chosen through field surveys and structural analyses. They are considered to be representative of the entire VSZ at different depths of exposure (shallowest conditions at Eyrs and deepest at Juval) and offer also the possibility to study and sampling the shear zone in
continuity, due to the good bedrock exposure. Each transect (Fig. 2) has been mapped at a 1:2,000 scale, identifying the distribution of protomylonites, mylonites and ultramylonites following the classification by Simpson and De Paor (1993). Sampling sites were accurately selected on the base of their representativity along the transects and the possibility they offered to perform microstructural and geochronological analyses.

**3.1.1. The Juval transect**

The easternmost transect is completely within the Texel unit. The cross section extends SE-NW, from the bottom of the Vinschgau valley to the Juval Castle. Here the road is entirely excavated in the bedrock, offering a continuous exposure of the entire shear zone. The bedrock mainly consists of granitoid orthogneiss (Partschinser orthogneiss) showing different mylonitization degrees. This transect almost corresponds to the westernmost termination of the Texel unit (Fig. 2). Besides
the Partschinser orthogneiss, the Texel unit here consists of garnet, staurolite and kyanite bearing paragneiss that are also affected by mylonitization. They chiefly occur in the upper and central part of the transect, alternating with the orthogneiss. Some amphibolite boudins (Fig. 3d) are also exposed within the paragneiss along the road that takes to the Juval Castle. Paragneiss display a decrease in grain size with respect to the ones outside the VSZ, especially in the central part of the transect, where they can be classified as protomylonites. Analysis of the shear strain distribution highlights a
symmetric increase, from rims to core, with ultramylonites and mylonites (Fig. 3) concentrated in the central part of the shear zone, whereas protomylonitic textures mainly occur both close to the structurally higher and lower margins. The mylonitic foliation is defined by the SPO (Shape Preferred Orientation, Passchier and Trouw, 2005) of biotite. The mylonitic lineation visible in outcrops mainly consists of elongated quartz aggregates and aligned biotite crystals (see section 3.2 for details). Foliation dips towards NNW, with mylonitic lineations that are nearly horizontal trending ENE-
WSW (Fig. 2b). Moving toward the core of the shear zone, the clast-matrix ratio decreases, as does the size of K-feldspar porphyroclasts (Fig. 3). Ultramylonites appear as dark-grey bands within mylonites, a few centimetres up to 3-4 metres in thickness (Fig. 3c). Quartz ribbons, a few millimetres in thickness and up to several decimetres in extension, frequently occur (Fig. 3e). Porphyroclasts are scarce within the ultramylonites, with a mean size not exceeding a few millimetres, whereas they commonly display a mean size of 20-30 mm outside the shear zone. Kinematic indicators represented by $\sigma$
and $\delta$ clasts (Fig. 3b), antithetic bookshelves in K-feldspar porphyroclasts and SCC' fabric, invariably point to a top-to-W/WNW sense of shear.

**3.1.2. The Schlanders transect**

The Schlanders transect is located about 20 km to the W of the Juval section (Fig. 2). This cross section of the VSZ is
entirely developed within the Ötztal polymetamorphic basement, here chiefly consisting of granitoid orthogneiss and minor two-mica paragneiss.

The upper portion of the structural transect extends outside the VSZ, where orthogneiss still preserve their metamorphic regional foliation of Variscan age (Schmid and Haas, 1989; Hoinkes et al., 1999; Thöni, 1999) gently dipping to the E. The finite strain distribution visible along this transect is symmetric, as was the case in the Juval section. Finite strain
increases from the margins toward the core of the shear zone, with a decreasing grain size of both matrix and K-feldspar porphyroclasts. On the base of the matrix/clast ratio (Simpson and De Paor, 1993), the orthogneiss is protomylonitic. Only towards the core of the shear zone mylonitic bands occur, ranging in thickness from 10 to 50 centimetres. Ultramylonites, which are frquent along the Juval transect, here occur only as 2-10 centimetres thick bands, with a dark-grey colour and an intense grain-size reduction (Fig. 3f). The mylonitic foliation dips to NNW with a variable dip angle
(Fig. 2b). Dip variations result from the occurrence of late stage S-vergent folds with E-W trending hinges that refold the



mylonitic foliation. The lineation associated to top-to-W shearing is nearly sub-horizontal or gently dipping to W (Fig. 2b), identified in the field by elongate quartz aggregates and white micas.

**Figure 3: Field photographs of Juval, Schlanders and Eyrs outcrops. (a) Proto-mylonitic Partschinser Orthogneiss with a**
**mylonite band, recognizable from the darker colour, reduced grain size and rare presence of K-feldspar porphyroclasts (SW**





**of Juval transect). (b) Proto-mylonite with asymmetric K-feldspar porphyroclasts showing a top-to-W sense of shear (S of Juval transect). (c) Mylonitic orthogneiss with ultra-mylonitic bands, recognizable from the dark grey-black colour, extremely reduced grain size and quartz ribbons presence (SW of Juval transect). (d) Mylonitic paragneiss with an amphibolitic boudin (central part of Juval transect). (e) Mylonitic orthogneiss with quartz boudins (E of Schlanders transect). (f) Mylonitic**
**orthogneiss with black, fine-grained ultra-mylonitic bands (E of Schlanders transect). (g) Proto-mylonitic orthogneiss (NW of Eyrs transect). (h) Phyllonite with quartz boudins (S of Eyrs transect). The image in the left corner shows the sample ERY-11, in which the calcite is clearly recognizable.**

### 3.1.3. The Eyrs transect

The Eyrs transect covers the W part of the exposed VSZ, close to the Schlinig Normal Fault (Fig. 2) that crosscuts the shear zone. The exposed part of the VSZ is here entirely developed again within the Ötztal polymetamorphic basement, chiefly made of granitoid orthogneiss. Protomylonites and mylonites are preserved only along the upper margin of the shear zone, whereas the remaining part, from 1300 down to 900 m.a.s.l., consists almost entirely of light grey to whitish phyllonites (Fig. 3g). The phyllonites' protolith is hardly identifiable in the field, but the widespread occurrence (see
section 3.2) of K-feldspar suggests that mylonites and ultramylonites developed on pre-existing granitoid orthogneisses. Phyllonites are also exposed on the right side of the Vinschgau Valley (Fig. 2), NW of Prad am Stilfserjoch. This part of the VSZ was probably the best known to past authors, and its fault rocks were previously known as the "Eyrs phyllites". The occurrence of dispersed carbonates (Fig. 3h) within the phyllonites led some authors to suppose the occurrence of Permian-Triassic carbonate sediments entrapped within the VSZ (Schmid and Haas, 1989).

Mylonitic foliation dips NNW with a variable dip angle due to the occurrence of the same S-vergent folds described in the Schlanders transect (Fig. 2b). The mylonitic lineation, here mainly identified by iso-oriented sericite crystal on the foliation planes, if far less evident than in other sectors of the VSZ, but a WNW-ESE trend (Fig. 2b) is generally recognizable. The SPO of chlorite, muscovite and quartz defines the mylonitic foliation. Extremely fine-grained quartz bands, a few millimetres thick, commonly occur (Fig. 3h). The light colour of the phyllonites turns frequently into a
brownish aspect (Fig. 3h). This is due to the occurrence of dispersed fine-grained carbonates (mainly calcite and Fe-dolomite, Fig. 3h) that likely originated from secondary fluids circulating along the shear zone. Secondary carbonates have been also found within orthogneiss-derived mylonites of the Juval transect.

### 3.2. Microstructural analyses and kinematic vorticity of flow

Field structural analyses served as a base for sampling of the different structural facies recognized along the VSZ studied transects. Samples were collected (Supplementary Table 1) all along the three transects at regular distances in order to obtain a complete representation of the whole exposed shear zone.

Granitoid orthogneiss, the main protolith of the mylonites of the VSZ, does not represent the best opportunity for *PT* estimates. The scarce changes in the equilibrium mineral assemblage and the variation in the mineral chemistry of
representative phases (white mica, biotite, K-feldspar and plagioclase) at greenschist and amphibolite facies conditions make difficult a precise estimate of pressure conditions. Maximum temperature of mylonites could instead be determined by the stability of chlorite *vs.* biotite along the mylonitic foliation and the dominant quartz recrystallization mechanism (Stipp et al., 2002).

In the Juval transect poorly deformed orthogneiss and proto- and mylonitic orthogneiss consist of plagioclase, quartz, K-
feldspar, muscovite, biotite and chlorite. Rutile, apatite, titanite and zircon occur as accessory phases. The regional foliation is made by the SPO of $Bt_{II} + Ms_{II}$ (Zantedeschi, 1991; Bargossi et al., 2010). Rare relics of $Bt_I$ have been reported on the easternmost termination of the Partschinser ortogneiss body, outside the VSZ (Bargossi et al., 2010). The mylonitic foliation is defined by the SPO of $Ms_{II} + Bt_{II}$, with white mica more abundant than biotite. Chlorite has been only rarely observed, mainly as an initial destabilization of biotite along rims of the $Bt_{III}$ grains.
In the poorly deformed orthogneiss, quartz is recrystallized mainly *via* bulging (BLG, Passchier & Trouw, 2005) recrystallization mechanism (Fig. 4a). Within proto- and mylonitic orthogneiss the dominant mechanism is subgrain rotation (SGR, Passchier and Trouw, 2005) recrystallization. A well-developed SCC' fabric (Fig. 4b and Fig. 5), together




with various groups of mica fish (group 1 to 5 of Passchier and Trouw, 2005; Fig. 4c) and asymmetric K-feldspar porphyroclasts (Fig. 4d) point to a top-to-W shear sense, as already observed in the field.

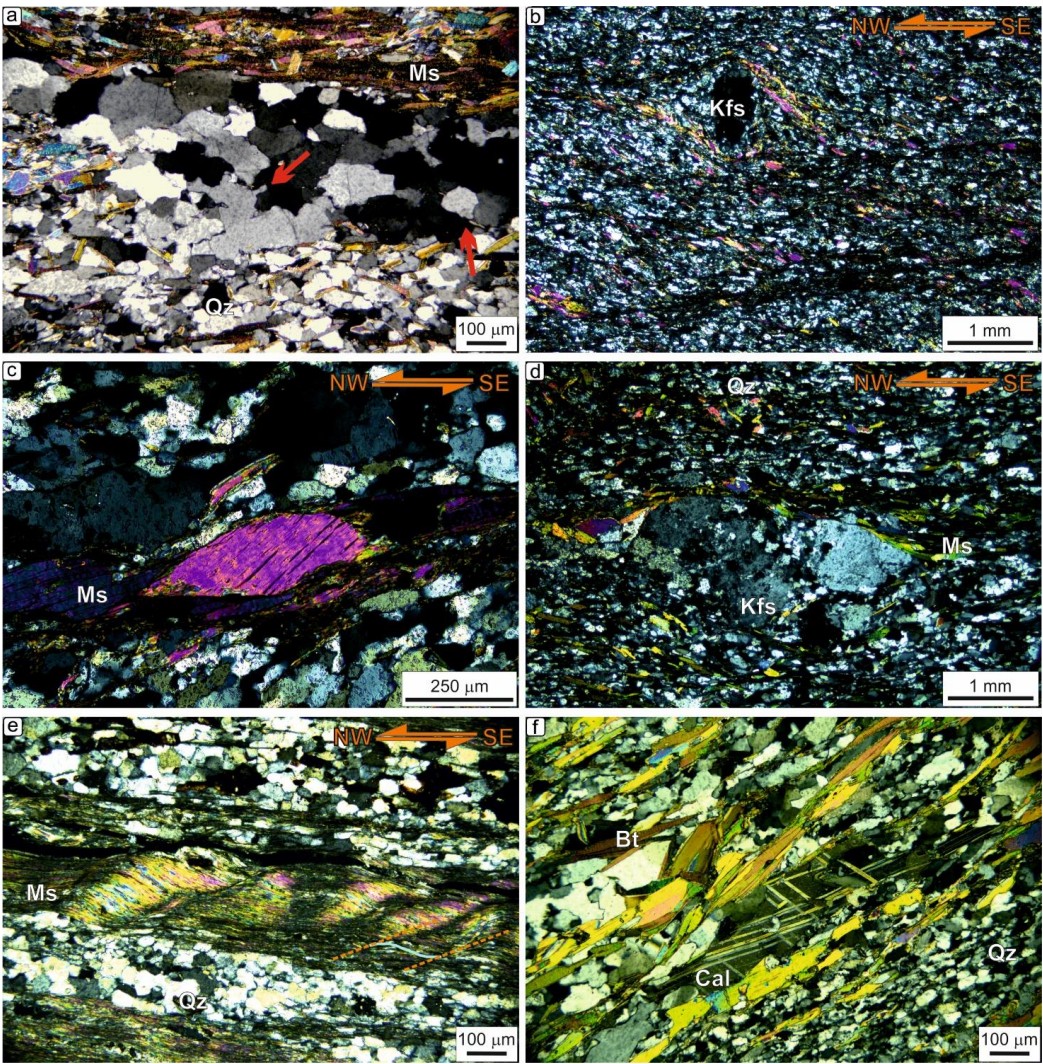

**Figure 4: Representative photomicrographs from the VSZ, crossed polars. (a) Quartz recrystallized *via* bulging recrystallization mechanism (red arrows) in poorly deformed Partschinser orthogneiss; (b) SCC' fabric together with asymmetric K-feldspar porphyroclasts point to a top-to-W shear sense; close up on (c) group 5 mica fish and (d) asymmetric K-feldspar porphyroclasts. (e) A well-developed crenulation cleavage in phyllonite, and (f) deformation twinning in calcite.**

In the Schlanders transect, chlorite substitutes biotite along the mylonitic foliation, pointing to lower temperatures during shearing. The syn-mylonitic foliation is defined here by the SPO of $Ms_{III}$ + Chl. The complete mineralogy of mylonites consists of plagioclase, quartz, K-feldspar, muscovite, chlorite and rare relics of biotite. Apatite, rutile and zircon are usually present as accessory phases. Mylonites along the Schlanders transect are often seen to grade into phyllonites, with a strong decrease in grain size and a progressive increase of muscovite with respect to chlorite. Within phyllonites, K-feldspar has been preserved only as relics of 1-3 mm in size, with muscovite becoming the major mineral phase. The occurrence of fine-grained muscovite domains characterize most of the phyllonites, often affected by a weak crenulation



(Fig. 4e) in response of E-W trending S-vergent folds, as previously described. Together with the increase of muscovite abundance, also carbonates increase. They occur as calcite and Fe-calcite crystals, reaching about 5% in volume of the whole rock. Quartz in mylonites display textures compatible with SGR recrystallization, whereas BLG is the only recrystallization mechanism observed in phyllonites.

In the Eyrs transect, the westernmost and, following Schmid and Haas (1989), the shallowest transect of the VSZ, phyllonites completely substitute mylonites. The whole of the shear zone consists of light-grey to whitish extremely fine-grained phyllonites. Phyllosilicates domains reach up to 60% of the rock volume, with muscovite as the major mineral phase. The abundance of calcite increase, reaching up to 10% in several outcrops. Calcite and Fe-calcite crystal show type 1 and type 2 deformation twinning (Ferrill et al., 2004; Fig. 4f). Quartz recrystallizes *via* BLG, no relics of SGR and GBM textures have been observed in samples from the Eyrs transect. SCC' fabric, foliation fish, mica fish (group 3 and 4) and K-feldspar porphyroclasts occur as kinematic indicators, pointing to a top-to-W sense of shear.

To define the type of flow within the VSZ, kinematic vorticity analyses were performed on five samples (JVL-16, JVL-12, JVL-11, JVL-6, JVL-14) collected along the Juval transect oriented perpendicular to the shear zone boundaries. The vorticity analysis were carried out on Juval samples, as they are suitable for vorticity estimates. Unfortunately, Schlanders and Eyrs mylonites are unsuitable for the application of any of the methods of vorticity estimates.

The analyses were performed on thin sections cut perpendicular to foliation and parallel to lineation (i.e. the XZ plane of the finite strain ellipsoid) using the C' shear band method (Kurz and Northrup, 2008), considering the mean value of the orientation of the synthetic shear bands following the interpretation of Gillam et al. (2013). Polar histograms used to derive the angle ($2v$) between the apophyses $A_1$ and $A_2$ are reported in Figure 5.

The C' shear band method reveal $W_m$ ranging from 0.64 up to 0.87, corresponding to a variation in simple shear component of 46 - 68%, with the kinematic vorticity decreasing towards the core of the shear zone.

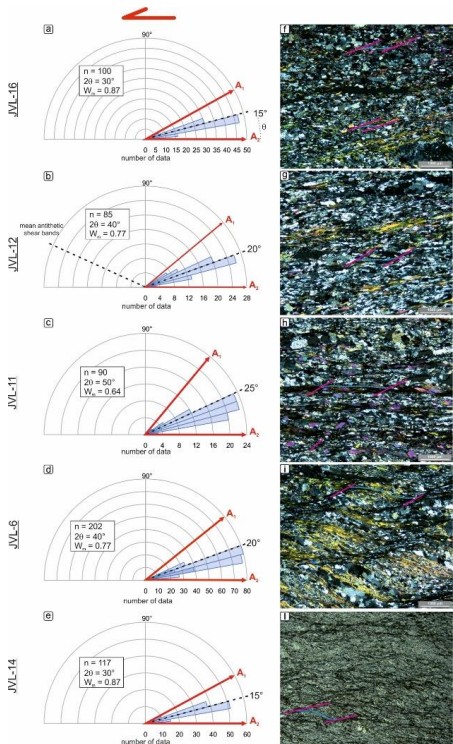

**Figure 5: Vorticity estimates through SC' (a–e) and relative SC' fabric (f-l). Polar histograms used to derive the angle θ and to calculate kinematic vorticity with the SC′ method (Kurz and Northrup, 2008) for samples JVL-16 (a), JVL-12 (b), JVL-11 (c), JVL-6 (d) and JVL-14 (e). C' shear bands are highlighted with violet lines and the main foliation S in dotted blue lines (f-l).**





## 4. Petrochronology
### 4.1. Mineral chemistry

Electron microprobe analyses were carried out using a JEOL 8200 Super Probe EMP at the Dipartimento di Scienze della Terra "A. Desio", Università degli Studi di Milano. Quantitative chemical analyses were performed on carbon-coated petrographic thin sections. Data acquisition was performed using an accelerating voltage of 15 kV, a beam current of 5 nA with a spot size of 1 μm. Natural silicates and oxides were used as standards. Muscovite and biotite analyses were recalculated as atoms per formula unit based on 11 oxygens.

Quantitative chemical analyses were performed on a non-mylonitic orthogneiss sample (JVL-15), on mylonitic orthogneiss samples (JVL-1, JVL-7, JVL-13, SCH-4) and on a phyllonite sample (ERY-11), with a total of about 100 points. Chemical analyses are reported in Figure 6 and Supplementary Table 2.

### 4.1.1. Muscovite

Considering all six samples, some compositional variations around the muscovite-celadonite join can be observed, with Si ranging between 3.1 and 3.4 apfu and Al ranging between 2.17 and 2.71 apfu (Fig. Xa). Muscovite in ERY-11 is characterised by the highest Al/Si ratios, while the other samples show a gradually decreasing Al/Si ratios, with JVL-7 and SCH-4 being the lowest (Fig. 6a). Muscovite in JVL-1, JVL-7, JVL-13 and JVL-15 samples is characterised by a negative correlation between Al and Si content (Fig. 6a), ranging from Si content between 3.14 and 3.35 apfu and Al

content between 2.17 and 2.55 apfu. Muscovite from sample JVL-7 displays a compositional cluster characterised by high Si content (3.30-3.34 apfu) and low Al content (2.17-2.22 apfu), whereas muscovite from sample JVL-15 shows an opposite cluster with low Si (3.17-3.20 apfu) and high Al (2.46-2.49 apfu) content (Fig. 6a). No variations in Al and Si contents have been detected between first- and second-generation muscovite (Ms1 and Ms2, respectively).

The Na/(Na+K) ratio (Fig. 6b, Guidotti and Sassi, 2002) of muscovite in sample ERY-11 is higher (0.09-0.13) than in

sample SCH-4 (0.01-0.02) and in samples JVL-1, JVL-7, JVL-13 and JVL-15 (0.02-0.07). No variations of the Na/(Na+K) ratio were observed on first- and second-generation muscovite. Muscovite in JVL-1, JVL-7, JVL-13 and JVL-15 samples show a homogenous distribution of the Na/(Na+K) ratio (0.02-0.07), with Si content ranging between 3.14 and 3.34 apfu (Fig. 6b).

### 4.1.2. Biotite

Analyses were performed on biotite from five samples, since it was absent from sample SCH-4. In the sample ERY-11 the $X_{Mg}$ ratio ranges between 0.54 and 0.56 (Fig. 6c-d), whereas $X_{Mg}$ of the other samples shows a cluster at lower values (0.34-0.46). As in the case of muscovite analyses, no significant chemical variations have been observed between the two biotite generations (Bt1 and Bt2).

The Si content of biotite in sample ERY-11 is homogenous, while in samples JVL-1, JVL-7, JVL-13 and JVL-15 it shows a cluster characterised by Si content between 2.74 and 2.83 apfu (Fig. 6c). The Ti content in biotite from samples JVL-1, JVL-7, JVL-13 and JVL-15 is clustered at values between 0.08 and 0.17 apfu (Fig. Xd), apart from a biotite-2 from sample JVL-7 which shows a remarkably lower value (0.01 apfu), like sample ERY-11 (0.001-0.006 apfu).

The K content (Suppl. Table 2) reveals that in sample ERY-11, all the spot analyses yielded low, sub-stoichiometric K in

biotite-1 and biotite-2. These values, very close to the detection limit (ca. 0.006-0.010 apfu) pertain to chlorite, as also supported by the matching element sums below 96% for these analyses (Suppl. Table 2). Biotite from samples JVL-1, JVL-7, JVL-13 and JVL-15 is characterised by K content of 0.93-1.00 apfu, except for a biotite-2 from sample JVL-7, which again shows low K content (0.02 apfu). As in the case of sample ERY-11, the low K content and the element sums (*c.* 89%; Suppl. Table 2) confirm the presence of chlorite in the sample.



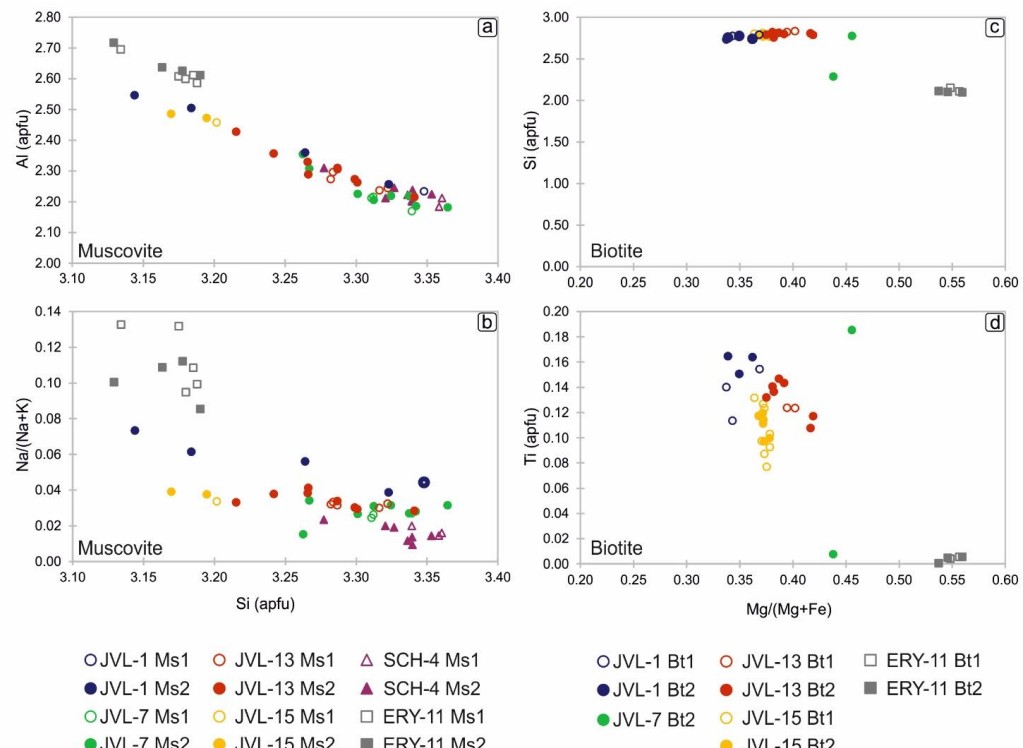

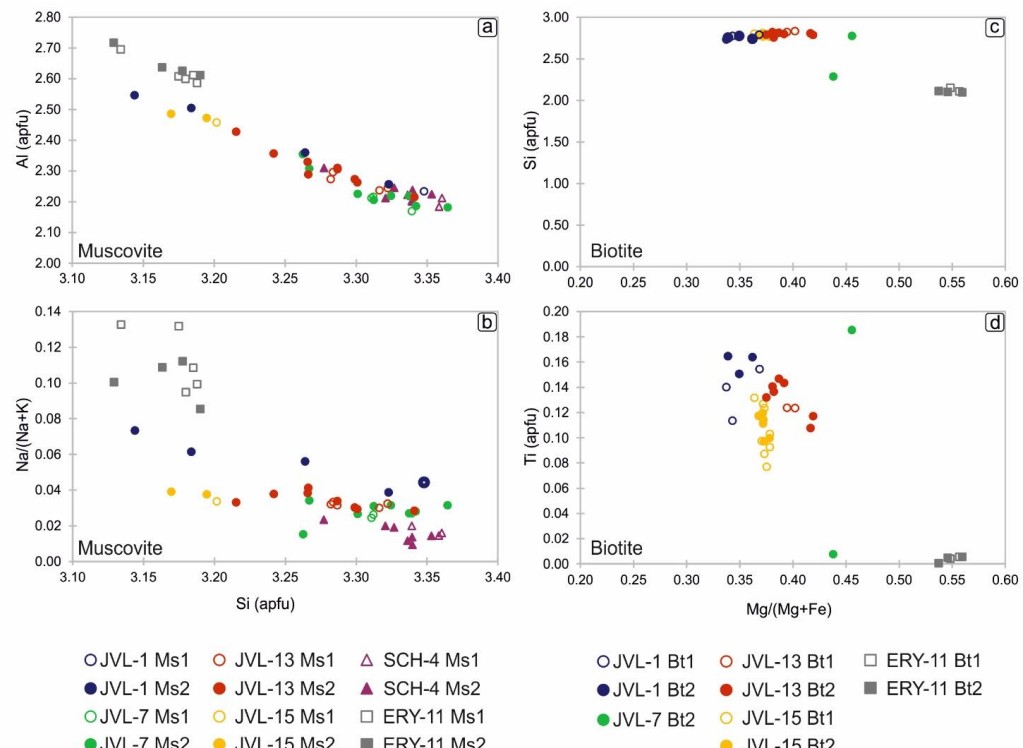

Figure 6: EPMA results showing compositional variation in muscovite (a–b) and biotite (c–d).

## 4.2. $^{40}Ar/^{39}Ar$ geochronological constraints on the VSZ activity

Muscovite and biotite used for $^{40}Ar/^{39}Ar$ geochronology were obtained from rock samples: JVL-1, JVL-7, JVL-13, JVL-15, SCH-1, SCH-4, SCH-5, ERY-3, ERY-8 and ERY-11. The mineral separation was performed at the Dipartimento di Scienze dell'Ambiente e della Terra, Università degli Studi di Milano - Bicocca. Rock samples were crushed and sieved, then muscovite and biotite in the 125-500 µm fraction were enriched by magnetic techniques and subsequently purified by hand-picking. Mica separates were cleaned and rinsed in pure deionized water by a two steps procedure in ultrasonic baths.

Mica samples were irradiated in the McMaster University Research Reactor (Hamilton, CA), carefully avoiding Cd shielding. $^{40}Ar/^{39}Ar$ step-heating analyses were carried out using a double-vacuum resistance furnace attached to a NuInstruments Noblesse rare gas mass spectrometer at the Dipartimento di Scienze dell'Ambiente e della Terra, Università degli Studi di Milano - Bicocca. The irradiation monitor was McClure Mountain hornblende (MMhb) with an assumed age of 523.98 ± 0.12 Ma (Schoene and Bowring, 2006). The decay constants were those of Steiger and Jäger (1977). Step heating experiments were conducted following the analytical protocols of Montemagni and Villa (2021). All $^{40}Ar/^{39}Ar$ data are reported in Supplementary Table 3.

### 4.2.1. The Juval transect

The age spectrum, the correlation diagram (Ca/K vs age) and the Differential Release Plots (DRP) for muscovite display similar features (Fig. 7). The $^{39}Ar$ degassing peaks in the DRP (occurring at temperature ranging in the ca. 1080 – 1300 K interval) correspond to the flat part of the spectrum and the lowest Ca/K values, as imposed by muscovite stoichiometry, which should be Ca-free. The heating steps showing high Ca/K and high Cl/K ratios (derived from the measured $^{37}Ar/^{39}Ar$ and $^{38}Ar/^{39}Ar$), monitor the degassing of Ca-rich alteration phases and have been disregarded in the age calculation. At





higher temperatures pertaining to muscovite or biotite degassing, the Ca/K and Cl/K reach a minimum. Biotite gas release patterns behave likewise, whereby the $^{39}$Ar-release peak occurs at lower temperature (ca. 960 K; Fig. 7 c-f-i-n), as extensively documented for Himalayan mylonites by Montemagni and Villa (2021) and Alpine ones in Montemagni and Zanchetta (2022). The age of mylonitic foliation on biotite, constraining the VSZ activity, is: 87.40 ± 1.06 Ma (JVL-1; Fig. 7a), 80.36 ± 0.46 Ma (JVL-7; Fig. 7d), 84.75 ± 0.40 Ma (JVL-13; Fig. 7g) and 92.28 ± 0.34 Ma (JVL-15; Fig. 7l).




**Figure 7: $^{40}$Ar/$^{39}$Ar age spectra, Ca/K vs. age diagrams and Differential Release Plots for Juval samples JVL-1 (a–c), JVL-7 (d–f), JVL-13 (g–i) and JVL-15 (l–n).**




### 4.2.2. The Schlanders transect

The $^{39}$Ar release patterns of the three muscovite samples SCH-1, SCH-4 and SCH-5 (Fig. 8) are different from those of Juval. All three Schlanders samples show a bimodal $^{39}$Ar peak pattern: one release around 940 K, typical of chlorite (see Montemagni and Villa, 2021), and one around 1050 - 1150 K, typical of muscovite. In SCH-4 and SCH-5 the muscovite

release peak corresponds to the lowest Ca/K ratios, i.e. the flat part of the spectra. In SCH-1 the muscovite release peak is not very evident but the three steps between 1080 and 1200 K correspond to low and homogenous Ca/K ratios, coherent with the Ar release in SCH-4 and SCH-5. The age of mylonitic foliation has been constrained at 95.35 ± 1.50 Ma (SCH-1; Fig. 8a), 89.37 ± 0.24 Ma (SCH-4; Fig. 8d) and 97.86 ± 0.24 Ma (SCH-5; Fig. 8g).

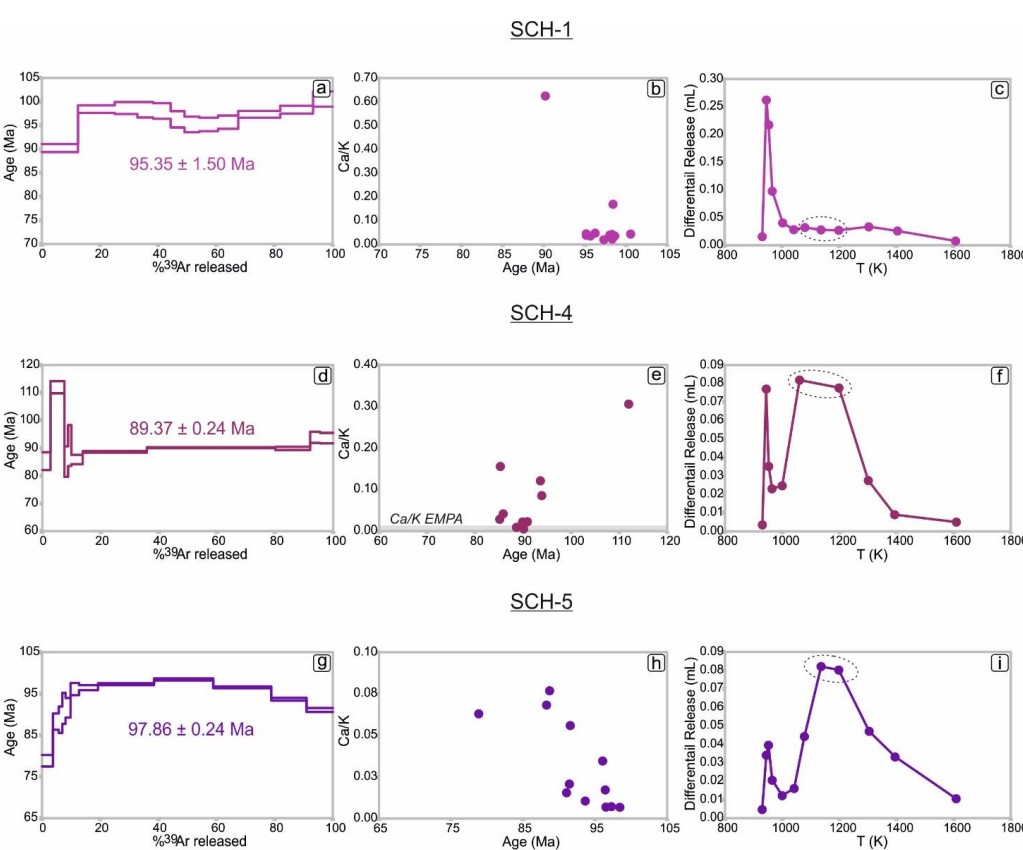


**Figure 8: $^{40}$Ar/$^{39}$Ar age spectra, Ca/K vs. age diagrams and Differential Release Plots for Schlanders samples SCH-1 (a–c), SCH-4 (d–f) and SCH-5 (g–i).**

### 4.2.3. The Eyrs transect

The phyllonites (ERY-3 and ERY-8) contain dispersed carbonates, evidence of massive fluid circulation (Fig. 3h). Moreover, due to the lower strain rate and/or lower temperature, the micas did not fully recrystallize during the Cretaceous faulting and give meaningless mixed ages with substantial Ar inheritance and muscovite ages are geologically meaningless (Fig. 9 a-d). The age of mylonitic foliation has been constrained to be 92.58 ± 1.56 Ma (ERY-11; Fig. 9g).




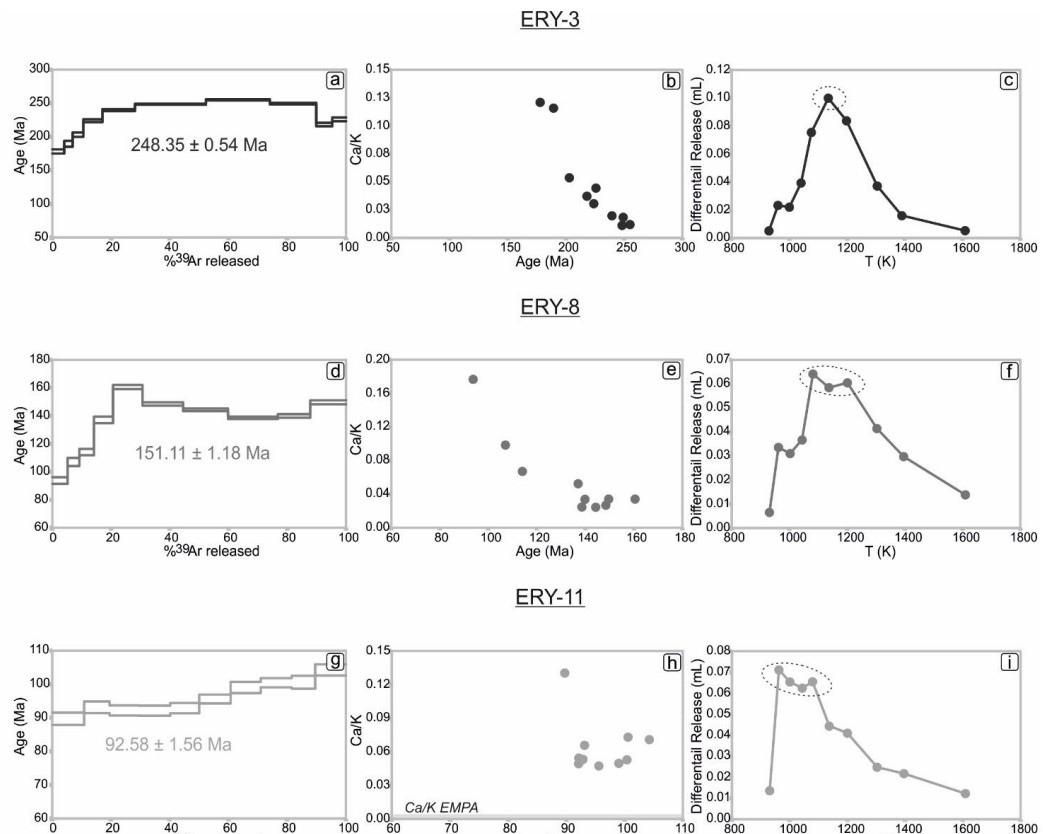

**Figure 9:** $^{40}$Ar/$^{39}$Ar age spectra, Ca/K vs. age diagrams and Differential Release Plots for Eyrs samples ERY-3 (a–c), ERY-8 (d–f) and ERY-11 (g–i).


## 5. Discussion

### 5.1. Kinematic and chronological evolution of the VSZ

Although the Vinschgau Shear Zone (VSZ) is likely the largest thrust-sense shear zone exposed in the Alps, no age constraints existed on the shearing activity of this huge intra-Austroalpine thrust, with its Late Cretaceous age only

inferred on the base of indirect evidence (Schmid and Haas, 1989) and a single Rb-Sr whole rock age of a deformed pegmatite (Thöni, 1986). Our $^{40}$Ar/$^{39}$Ar data are the first attempt to constrain the time interval of activity of the VSZ, combining geochronological results obtained across the shear zone and along its dip direction.

The evolution of shear zones has been deeply investigated in terms of spatial variation especially concerning their length and thickness (Hull, 1988; Mitra, 1992; Means, 1995; Vitale and Mazzoli, 2008, 2010; Fossen, 2010; Fossen and

Cavalcante, 2017). This effort was aimed at defining the parameters that may influence the evolution of one type of shear zone with respect to another one. If the growth in length of a shear zone is essentially due to linkage of different branches forming a composite system of shear zones (Fossen and Cavalcante, 2017), the growth in thickness may be influenced by different mechanisms. Four ideal models of shear zone evolution have been proposed and discriminated based on shear strain gradient, kinematic vorticity and plane or triaxial strain (Vitale and Mazzoli, 2008; Fossen and Cavalcante, 2017).

Processes of strain hardening or strain softening promote the thickening and the thinning of the shear zone, respectively *type-1* or *type-2* models. In the *type-1* model, the deformation concentrates in the margins of the shear zone, leaving inactive the inner portion. On the contrary, in the *type-2* model the deformation shifts and concentrates in the inner portion



of the shear zone as strain accumulates, leaving the margins inactive. In addition, *type-3* model is related to a strain
weakening process, even if its active thickness remains constant with time. *Type-4* model expands in thickness but, unlike
*type-1*, all the thickness remains active through time. According to the several models proposed for shear zone evolution
(Fossen and Cavalcante, 2017 with references), the VSZ followed a *type-2* evolutionary model, with increasing finite
strain from margins to the core of the shear zone (Fig. 10). This pattern of finite strain distribution is demonstrated for the
VSZ by the occurrence of ultramylonites at the core of the Juval transect (Fig. 10), whereas protomylonites derived from
the Partschinser orthogneiss are preserved only at the margins. The kinematic vorticity of flow follows the same
symmetric distribution, with $W_m$ that increases from the rims to the core (ca. 0.64 at margins and ca. 0.87 at the core, Fig.
10). The patterns displayed by the shear strain points to a strain-softening behavior of the shear zone (e.g. Vitale and
Mazzoli, 2008). At the microscale, the processes that may cause strain softening are: (i) recrystallization during shearing,
especially when associated with subgrain rotation recrystallization mechanism (Passchier and Trouw, 2005), which lead
to formation of new grains with lower dislocation density that results in a deformation at lower differential stress; (ii)
shear bands fabric formation (Fig. 5), as shear bands imply local grain-size reduction, a mechanism which is invoked in
low-to-medium mylonitization to promote strain weakening (Fossen and Cavalcante, 2017 and references therein). Shape
preferred orientation (Fig. 4) may also weakens the rheology as recrystallized or newly formed metamorphic minerals,
mainly micas and quartz in these lithologies, have intracrystalline slip planes aligned in the shear direction resulting in a
favorable arrangement to be easily deformed (Passchier and Trouw, 2005 with references).


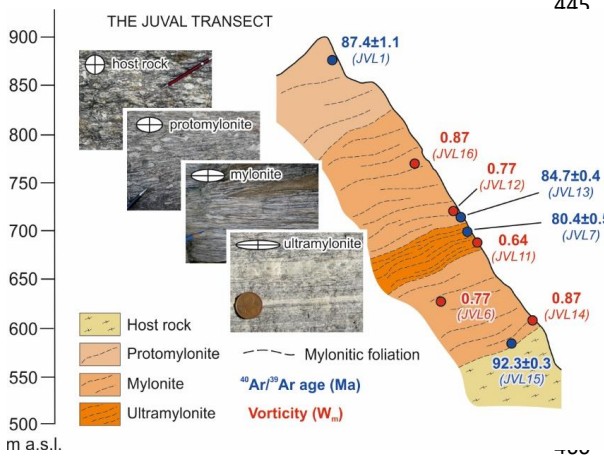

Figure 10: schematic cross section of Juval transect
showing the occurrence of protomylonites,
mylonites and ultramylonites developed along the
strain gradient from rim to core of the shear zone,
with $^{40}Ar/^{39}Ar$ age (in blue) and kinematic vorticity
number (in red). Samples not on the profile have
been projected along the strike of the mylonitic
foliation.

The VSZ deepens from W to E, as suggested by previous workers (Schmid and Haas, 1989) and confirmed by the present
data, with muscovite-chlorite phyllonites overprinting muscovite-biotite mylonites in the western part of the shear zone.
Broadly, the $^{40}Ar/^{39}Ar$ ages rejuvenate from W to E, opposite to the transport direction and consequent exhumation of the
Austroalpine units in the hanging wall. This age pattern points for younger ages of shearing in the E (i.e. Juval area) where
deeper portions were still being deformed (Fig. 11), while in the W the shear zone was already at shallow crustal depths.
Beyond the deformational path of the VSZ, which clearly follows a strain softening *type-2* shear zone evolution model,
the novelty of our work is the integration of microstructural and kinematic analyses with age profiling of the shear zone
along its transport direction, along depth, and across strike (Fig. 11). The obtained age pattern allows the reconstruction
of a time-resolved evolution of the shear zone during its progressive activity and exhumation.

The $^{40}Ar/^{39}Ar$ ages recorded by micas in the Schlanders and Juval transects reveal a clear younging trend from the margin
to the inner zone of the shear zone (Figs. 7, 8 and 11). This trend of younging mica ages is paralleled by increasing finite
strain and decreasing simple shear component (Fig. 11). The correlation between ages and deformation suggests a
progressive migration of deformation towards the inner portion of the VSZ, with the margin becoming inactive (as



testified by older mylonite ages ≥ 97 Ma), while deformation concentrates in the middle of the shear zone where micas show younger ages down to 80 Ma.

### 5.2. The VSZ in an evolving orogenic wedge

The evolution of the Eo-Alpine orogenic wedge of the Eastern Alps is generally related to the closure of the Meliata-Hallstatt ocean, located in an intra-Austroalpine position (e.g. Schmid et al., 2004), or separating the former Austroalpine and Southalpine domain (e.g. Neubauer et al., 2000). Other interpretations consider instead the possibility that the entire Austroalpine orogenic wedge formed in the Late Cretaceous in a pre-collisional setting (Zanchetta et al., 2012, 2015). Irrespective of the geodynamic scenario of the Eo-Alpine orogen in the Eastern Alps, the VSZ acted as a crustal-scale shear zone promoting nappe stacking and exhumation within an orogenic wedge chiefly made of continent-derived tectonic units. The ages of shearing along the VSZ indicate that the shear zone was already active at 97 Ma (Fig. 7, Fig. 8 and Fig. 11), at least 7-8 Ma before the pressure peak recorded by the Texel eclogites and the amphibolitic peak in the Schneeberg unit. The ages of the VSZ in the Schlanders transects overlap with published/available ages related to the peak of Alpine metamorphism in the Ötztal basement, suggesting that thrusting within the Austroalpine domain started where units in the hanging wall of the VSZ had already reached (Ötztal) or were close to the metamorphic peak (Texel and Schneeberg units). These ages overlap between shearing and metamorphic peak is explained by a rapid exhumation of these HP rocks *via* thrusting in the Eo-alpine orogenic wedge. Handy et al. (2010) argued that the Eo-alpine orogen, in the time span 118-84 Ma, was subjected to intracontinental subduction and basement nappe stacking, whose evidence is the E-W trending belt of Late Cretaceous, HP rocks of the Koralpe-Wölz nappe complex (Schmid et al., 2004; Thöni et al., 2008), to which the Texel and Schneeberg units belong. The exhumation of these HP rocks was both westward and northward (Handy et al., 2010).

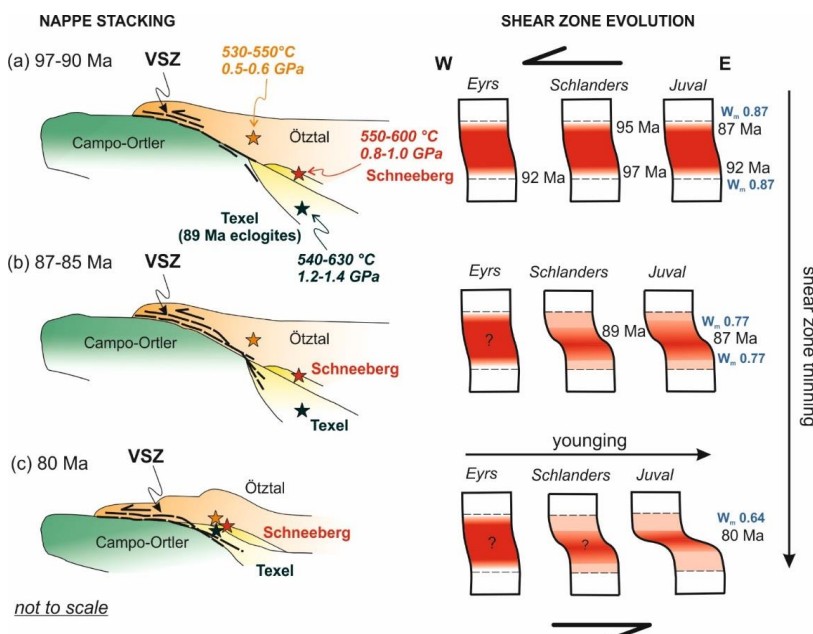

**Figure 11: Evolution of the Vinschgau Shear Zone in the orogenic Eo-alpine wedge during the Late Cretaceous. Activation of the VSZ (a) in the time span 97-90 Ma affecting the Ötzal unit, (b) in 87-85 Ma the VSZ affected also the Texel unit and (c) during the late stage of shearing exhumed the Texel and Schneeberg units to Ötzal *PT* conditions. In (a) the *PT* conditions for Ötzal, Texel and Schneeberg units are reported. The evolution of the VSZ in the studied transects in term of ages, finite strain and kinematic vorticity is also shown.**



The W/SW-directed Late Cretaceous thrusting (Eisbacher and Brandner, 1996; Ratschbacher, 1986; Schmid and Haas, 1989; Viola et al., 2003; Handy et al., 2010) and consequent nappe stacking under pressure-dominated, amphibolite to greenschist facies conditions, is unusual, if compared to major N/NW-directed direction of transport for other sector of Alps. This scenario has been explained through a complex microplate configuration at 94 Ma, which necessarily invokes the eastward faster accommodation of the Iberia microplate with respect to the Adria microplate, also moving eastward with respect to Europa, but at lower rates (Handy et al., 2010).

The age of the pressure metamorphic peak within HP units has been reported between 95 and 89 Ma (Thöni, 2002; Habler et al., 2006; Miller et al., 2005; Thöni et al., 2008; Janák et al., 2009; Zanchetta et al., 2013), with partially overlapping ages related to their rapid exhumation (e.g. Fügenschuh et al., 1997; Sölva et al., 2005).

        As the Texel and the Ötztal units are now in tectonic contact at the eastern termination of the VSZ, at least 20 km of vertical exhumation of the Texel unit should have been accommodated along the VSZ, considering the difference in peak

metamorphic pressure recorded by the two units (Fig. 11a): 1.2-1.4 GPa for the Texel unit (Habler et al., 2006) and 0.5-0.6 GPa for the Ötztal basement (Purtscheller and Rammlmair, 1982). Considering a lithostatic pressure gradient of 0.03 GPa/km and a consequent difference in terms of pressure of 0.6 GPa between the Ötztal and the Texel unit, such vertical displacement corresponds to a minimum of about 40 km along the shear zone, taking into account a dip of about 30°, which is typical of thrust-sense shear zone active through crustal depths controlling the exhumation of high-grade units

in their hanging wall (e.g. Jamieson et al., 2004). The amount of E-W shortening within the Austroalpine basement nappes has been estimated to be ≥100–150 km (Manatschal and Bernoulli, 1999; Schmid et al., 2020).

        The VSZ yields ages in the range of 80 Ma (the youngest age at the inner portion of Juval transect) and 97 Ma (the oldest age in the Schlander area) almost coeval with the peak metamorphic age in the Ötztal and Texel units at ca. 85-90 Ma (Zanchetta et al., 2013, Fig. 11). Hence, we argue that the VSZ has been active at least for 17 Ma, promoting the rapid

exhumation of the Texel eclogite in the Late Cretaceous. Syn-shearing exhumation of the Texel and Schneeberg units continued at least up to 76 Ma, as testified by the age of greenschist facies mylonites along shear zones within these two units (Sölva et al., 2005). This scenario was depicted also by Handy et al. (2010), who suggested a rapid exhumation of the Koralpe-Wölz units during the Cenomanian - Santonian (ca. 94-84 Ma) when the Gosau Group (syn- to post-orogenic clastic sediments sometimes deposited in intra-orogenic extensional basins) sealed thrusts in the Austroalpine basement

and in the Northern Calcareous Alps.

        Exhumation models for crystalline rocks have been extensively proposed, modeled and tested for the Himalayan orogen, where the exhumation of the metamorphic core of the belt was matter of debate in the literature (e.g. Montomoli et al., 2015; Carosi et al., 2018; Montemagni, 2020). Regardless the specific model, the scientific community agrees that exhumation in the Himalayas has been driven by two opposite shear zone bounding the crystalline core: a normal-sense

at its top and a thrust-sense shear zone at its bottom, respectively (e.g. Godin et al., 2006; Montomoli et al., 2013 for reviews).

        In this perspective, we argue that the VSZ played a key role in the exhumation of HP rocks in the Eastern Alps, and we can speculate that the units containing HP rocks have been exhumed by a (non-necessarily coeval) shearing of a thrust-sense shear zone at the base and a normal-sense shear zone at the top of the orogenic wedge, i.e. above the Ötztal-Stübai

complex wedge, later eroded in post-Cretaceous times.

        Taken together, all these data indicate that the VSZ had a long-lasting evolution of at least 17 Ma. The minimum accommodated displacement of ca. 40 km implies a displacement rate of 2-2.5 mm/yr that is significantly higher than values reported in Vitale and Mazzoli (2008) that are of 0.10 mm/yr for thinning shear zone (*type-2*). The accommodated displacement of the VSZ is coherent with slip rate values of ca. 10 mm/yr reported by Stübner et al. (2013) for an

intrabasement normal sense shear zone in the Pamir plateau. The exhumation continued rapidly as suggested by thermochronological data in the eastern Ötztal-Stübai complex unit where cooling below 60 °C at 60 Ma has been reported (Fügenschuh et al., 1997).



### 6. Conclusions

The VSZ is one of the prominent intra-basement thrust-sense shear zone developed in the Alps, promoting the exhumation of HP rocks within the Eo-alpine orogenic wedge.

Our approach fully constrains its kinematic and temporal evolution:

(i) Finite strain and kinematic vorticity values reveal an evolution compatible with a *type-2* thinning shear zone.

(ii) $^{40}Ar/^{39}Ar$ geochronology defines the shearing activity to be comprised between 97 and 80 Ma, resulting in
a long-lasting deformation history.

(iii) The novelty of our work is the combination of microstructural and kinematic analyses with age profiling of the shear zone both along its transport direction and across strike.

*Author contribution.* SZ, AZ and CM designed the study, CM and MR carried it out. The paper was prepared by CM and SZ and revised by AZ – with the contribution of all co-authors. All authors participated in fieldwork and in the various scientific discussions.

*Competing interests.* None of the authors has any competing interests.

*Acknowledgements.* This research was funded by MUR through the PRIN project FAST, 2021-NAZ-0299 (CUP: J33C22000170001), and by the CARG project of the Autonomous Province of Bozen – South Tyrol. We thank A. Risplendente (Università degli Studi di Milano) for his support during the electron microprobe analyses and V. Barberini (Università degli Studi di Milano - Bicocca) for the maintenance of the mass spectrometer.

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

**Supplementary Table 1: sample list with coordinates in UTM32N and elevation m a.s.l.**

**Supplementary Table 2: Electron probe micro analyses of micas (wt% oxides). Atomic proportions (apfu) were recalculated based on 11 oxygens.**

**Supplementary Table 3: $^{40}$Ar/$^{39}$Ar data. All Ar isotope concentrations are given as mL. K, Ca and Cl concentrations are calculated from the sample mass and the $^{39}$Ar, $^{37}$Ar and $^{38}$Ar concentrations, respectively.**