# Peer review of "Kinematics and time-resolved evolution of the main thrust-sense shear zone in the Eo-Alpine orogenic wedge (the Vinschgau Shear Zone, Eastern Alps)"

_EGUsphere, 2023_

## Author Response (AR1)

Dipartimento di Scienze dell'Ambiente e della Terra

*Piazza della Scienza 4*

*I-20126 Milano*

**Manuscript no. egusphere-2023-126**

Dear Editor,

please find attached the revised version of our manuscript:

 "**Kinematics and time-resolved evolution of the main thrust-sense shear zone in the Eo-Alpine orogenic wedge (the Vinschgau Shear Zone, Eastern Alps)**"

authored by

*Chiara Montemagni, Stefano Zanchetta, Martina Rocca, Igor M. Villa, Corrado Morelli, Volkmar Mair and Andrea Zanchi*

We wish to thank the reviewers Paolo Conti and Frantz Neubauer for their suggestions and careful comments which helped us to improve the paper quality.

In the following notes we account and answer to all the reviewers' comments, which have been accepted and incorporated in the text. We carefully checked the text for grammar and spelling mistakes and made all the requested reviewers' corrections. All changes made to the text are highlighted in the "tracked changes" version of the manuscript and lines number in our replies refers to the "tracked changes" file.

We also modified the figures in order to satisfy all reviewers' requests.

We hope that the revised manuscript will be judged improved and will be suitable for publication in Solid Earth.

Sincerely yours

Chiara Montemagni

Milano, 30 March 2023

**Changes to figures**

Figures have been modified according to reviewers' suggestions. The main changes are reported below.

**Figure 1:** we amended the misfit of colour between legend and map for "Calcschists with ophiolites" and "Ortler-Campo Complex", we added the Naif Thrust in the map and the basement unit in the footwall of the VSZ immediately to the E of the Glurns Fault has been attributed to the Campo-Ortler nappe, with the Sesvenna basement that underlies only the Engadine Dolomites.

**Figure 2:** we increased the size of Lm and poles to Sm in lower hemisphere diagrams and added some foliation and lineation attitudes in the map, inside and outside the VSZ.

**Figure 3:** we added field orientations to all photos and enlarged inset in fig 3h.

**Figure 4:** we amended photos (c) and (e), we change photo (b) inserting a photo showing static Wm ($Wm_2$); we added sample nos. on the photos.

**Figure 6:** we amended Ms with Wm.

**Figure 7:** we wrote sample nos. inside diagrams and used the same scale between (a) and (b).

**Figure 8:** we wrote sample nos. inside diagrams.

**Figure 9:** we wrote sample nos. inside diagrams.

**Reviewer#1 Paolo Conti**

**Reviewer***: The paper is very interesting, clear and well written, and can be published with only minor revisions. The figures are ok (fig 4c and 4e must be corrected). I have some concern using the term "finite strain" in such rocks: grain size and shape of grains we observe today in rocks has nothing to do with finite strain suffered by rocks, but are only relate to deformation mechanisms and syntectonic (dynamic) recrystallization. More comments and corrections in the attached PDF.*

**Reply to specific comments in the annotated pdf.**

**Line 184-186:** We have modified "finite strain" with "cumulative shear strain". It is right that we cannot quantify the finite strain, here however we intend a qualitative variation and increase of shear strain from the host rock toward the core of the shear zone.
**Line 236:** Yes, we used $Bt_1$ and $Wm_1$ for first generation and $Bt_2$ and $Wm_2$ for second generation: we have specified that at lines 340-343. We changed the format of numbers indicating the generation of minerals from roman to arabic format, as labels that are easier to read in the figures.
**Line 441-442:** We have corrected SPO with LPO.
**Line 472-473:** We did not observed any change in the deformation mechanisms; we measured the simple shear percentage through vorticity estimates.
**Line 539-540:** No, there is no evidence of a normal sense shear zone atop of the Otztal-Stubai Complex in the Vinschgau valley area. The remnants of this shear zone could be identified with the tectonic contact between the Ötztal-Stubai unit and the Steinacher Decke, west of the Brenner Pass, please see the replay to Reviewer#2 comments that raised the same problem and text added at lines 582-588 of the manuscript revised version.
**Line 553:** We have modified "finite strain" with "cumulative shear strain", we did not observed any change in the deformation mechanisms.
**Figure 11:** The dashed lines represent the thickness and length of the shear zone. However, we added this statement in the caption of the figure to be clearer. The increase in strain in the Juval profile with respect to the Eyrs profile is due to a not a coherent behaviour of the Ötzal unit during deformation.
**Line 580-581:** We have amended the citation.

**Reviewer#2 Frantz Neubauer**

**Reviewer**: *This is a very interesting manuscript documenting kinematics, white mica and biotite composition and corresponding $^{40}Ar/^{39}Ar$ ages from the Vinschgau Shear Zone in Eastern Alps and deduce a reasonable model. The Vinschgau shear zone is clearly one of the largest and most important eo-Alpine (Cretaceous) shear zones in the Austroalpine unit of Eastern Alps. This shear zone has a high importance because it exhumes also the eo-Alpine Texel eclogites in the hangingwall unit, for which a SHRIMP U-Pb age of $86 \pm 5$ Ma. The mixed structural and geochronological approach applied in three sections allow to document that the shear zone was active in an increasingly shallow crustal level. The ages vary over a relatively wide range between 97 and 80 Ma indicating that the western part with older ages was exhumed first. The manuscript represents a significant progress for tectonics in the western Austroalpine units and also includes methodological progress because of the combination of structural and geochronological methods.*

*To reach full clarity on the significance of the new data, I propose to pay attention to following points during revision:*

*1) Mention more clearly the only weak overprint within greenschist facies in the Campo-Ortler-Schesvenna units in the footwall, from which some old K-Ar and Rb-Sr whole-rock-white mica ages were published by Thöni 1981.*
**Reply**: We have mentioned this at lines 45-46.

**Reviewer**: *2) Use of "muscovite": Personally, I prefer "white mica" because of two reasons: In the study, the EPMA analyses show a broad range of compositions, some have a relative high phengite component and are rather phengites/phengitic muscovites. The second problem is the quite common admixture of a paragonite component to white mica in greenschist facies rocks.*
**Reply**: we corrected the term "muscovite" with "white mica" in the whole manuscript, including figures.

**Reviewer**: *3) Figs. 7 - 9: It remains unclear which ages are actually listed in the Ar release patterns/age spectra on the left side. What means the stippled circIes in the diagrammes on the right side (T vs. Differential Release). Is there any information, from which mineral phase the Ca in all these patterns is coming from? Did you try calculate plateau and/or isotope inversion ages? The authors refer to their study of Himalayan shear zones (Montemagni & Villa, 2001), but this study is partly diferent by using mineral mixtures (e.g., white mica and chlorite).*
**Reply**: We added (lines 394-397) a sentence to clarify which ages are reported in the figures and to clarify that our approach takes into account the information deriving from the age spectra, the Ca/K ratios and the differential $^{39}Ar$ release. Together all these information allow obtaining isochemical ages. The observation is that the phases being degassed in the different steps are different (for a more detailed discussion, see Villa, 2021). The identification of the near-stoichiometric steps as pertaining to the degassing of white mica is straightforward. In contrast, the alteration phases are generally smaller than 1 µm and can only be seen as subtle colour contrast by EPMA but not quantitatively determined. Any isochron that mixes white mica and alteration phases, which are by definition non-cogenetic and non-isochronous, is certainly incorrect.
The alteration phases may give both younger and older apparent step ages; what is important is the observation that alteration phases have a non-stoichiometric Ca/Cl/K signature.
The study of Montemagni and Villa (2021) considers mixtures of white mica, biotite and chlorite to verify the approach and to demonstrate the different $^{39}Ar$ release among them, but this approach is subsequently applied to separates of biotite and white mica in order to identify the steps to be selected for the ages calculation. These steps correspond to the lowest and homogeneous Ca/K ratios and highest $^{39}Ar$ release (the dashed circles in figures 7-8-9). We cleared this aspect in the text and in the captions.

**Reviewer**: *4) More documentation of representative microfabrics of dated samples is urgently needed. In the text, several times two generations of white mica and biotite are mentioned.*
**Reply**: we added a photo (Fig. 4b) in which the generations of $Wm_1$ and $Wm_2$ are clearly visible.

**Reviewer**: *5) Discuss the origin of the scatter of Ar-Ar ages in some studied sections, e.g., in Schlanders. Discuss also the different meaning of white mica and biotite ages. Could the scatter represent mixtures of variable composition as shown in EPMA data?*

**Reply**: The mica diachronism dates the diachronic strain focusing to the centre of the shear zone. In Schlanders and Eyrs samples, we dated white mica as biotite is almost completely retrogressed. In Juval Bt and Wm give the same age within analytical error and refer to the same deformation event.

**Specific remarks**

*42: Ratschbacher et al. 1991: I think the correct reference is Ratschbacher et al., 1989. Ratschbacher et al. 1991 discuss the Oligocene-Miocene extrusion. In Ratschbacher et al. (1989) eo-Alpine ductile structures are discussed.*

**Reply**: Yes, we amended the citation also in the references list.

*42-44: "the entire central Austroalpine nappe stack was affected by W-directed tectonic transport during the first stages of the Late Cretaceous Alpine deformations": I think this is not fully correct. In the Central Austroalpine nappe east of the Tauern window, thrust deformation deformation at ca. 100-88 Ma is overprinted by ESE-directed ductile normal faulting, mainly east of the Tauern window, but also by Schlinig fault and potentially the ductile shear zone at the structural base of the Steinach nappe.*

**Reply**: The reviewer is right. ESE directed extension is documented both along the Schlinig normal fault (e.g. Froitzheim et al., 1997) and close to the eastern termination of the Ötztal-Stubai complex (Fugenschuh et al., 1997). If the overprinting of tectonic structures in the Vinschgau valley is clear, the relationships with the VSZ and the top-to-ESE shear zone at the base of the Steinacher nappe are not so straightforward. Considering the jump in the metamorphic grade and age of metamorphism between the underlying Ötztal-Stubai complex (garnet amphibolites facies) and the quartz-phyllites (ca. 350°C of peak temperature, Lünsdorf et al., 2012) and pre-Permian metamorphic ages (Rockenschaub et al., 2003) of the Steinach nappe, a normal sense shear zone could be imagined between the Ötztal and the Steinacher crystalline, bringing in contact the Austroalpine units that escaped the Eo-Alpine metamorphism, with underlying ones that were instead affected by it. We addressed this topic at lines 582-588.

*45: Mention also the eo-Alpine (Cretaceous) age of the up to greenschist facies metamorphism of the Austroalpine units underneath the VSZ.*

**Reply**: We have mentioned this at lines 45-46.

*57-58: The Noric thrust (at the base of the Graywacke zone) is another huge important ductile thrust (Ratschbacher, 1986), and the first one studied by modern kinematic/structural methods.*

**Reply**: we added this statement (lines 61-62).

*85-86: Cite here also the recent paper on Schneeberg/Ötztal relationships: Klug & Froitzheim, 2022.*

**Reply**: Done

*123: There is no formal "Middle Cretaceous" in the International Stratigraphic Chart. Write middle with lower case.*

**Reply**: Yes, it is upper case as it is at the beginning of a sentence after a full stop. However we corrected to Mid in order to avoid misunderstanding.

*127: What is the metamorphic grade of the Matsch unit?*

**Reply**: Upper greenschist facies, we added this in the text (lines 130-132).

*L.231-232: Only a comment: If there is a reasonable metamorphic T estimate, the phengite barometer of Massonne and Schreyer (1987) could be applied to estimate pressure.*

**Reply**: the application of the Massonne and Schreyer (1987) barometer is not permissible, as our samples do not match the paragenesis for which the barometer was calibrated.

*248: "group 5 mica fish"?? Add: "accoding to the classification of Passchier and Trouw, 2005"*

**Reply**: done.

*273: Mention why these Eyrs mylonite samples were not suitable for vorticity estimates.*
**Reply**: we added (lines 322-323) why Eyrs and Schlanders mylonites are not suitable for vorticity estimates.

*345: Explain what is biotite-1 and biotite-2.*
**Reply**: we added a sentence (lines 340-343) to clarify this statement.

*506: I think it is necessary to distinguish the eo-Alpine Cretaceous deformation iin Austroalpine units from Austroalpine/Adria/Europe collision during Paleogene times.*
**Reply**: In this work we deal only with the Eo-Alpine evolution of the Austroalpine domain in the Eastern Alps. The main focus is on the evolution through time and space of the VSZ and the role of crustal-scale shear zones in the building of an orogenic wedge. Discussion on the Europe-Africa collision and the consequent indentation of Adria is beyond the scope of the paper, we have confined our geodynamic interpretation to the Late Cretaceous, i.e. the time interval during which the VSZ was active.

*533-535: Based on structural and geochronological data, a similar model was also proposed for the type locality eclogites in the Saualpe region (Wiesinger et al., 2006). A recent study (Schulz and Krause, 2021) documented the younger age in amphibolite-grade footwall units underlying the eclogite-bearing unit.*
**Reply**: Thank you for bringing this to our attention; we added this statement at lines 576-578.

*539-540: Mention also where this normal-sense shear zone is located, e.g., top of Ötztal or Ötztal/Texel boundary*
**Reply**: We addressed this topic at lines 582-588.

**Technical/editorial problems (see also few further remarks in the enclosed annotated pdf-file)**

*77: shear zone develops (singular)*
**Reply**: done.

*163: Correct: "The paragneiss displays" or "Paragneisses display"*
**Reply**: done.

*188: Correct: frquent*
**Reply**: done.

*208: Explain abbreviation a.s.l.*
**Reply**: done.

*237: Correct: ortogneiss*
**Reply**: done.

*248: "group 5 mica fish"?? Add according to the classification of Passchier and Trouw, 2005.*
**Reply**: done.

*251-252: Add: "lower temperatures during shearing than in the Juval section".*
**Reply**: done.

*278: correct to: method reveals*
**Reply**: done.

*321: explain abbreviation apfu for non-specialists*
**Reply**: done.

*342: Fig. Xd = (Fig. 6d)?*
**Reply**: Yes, it has been amended.

*345: Explain what is biotite-1 and biotite-2.*
**Reply**: done.

*438: either: "mechanisms (Passchier and Trouw, 2005), which lead" or "mechanism (Passchier and Trouw, 2005), which leads"*
**Reply**: done.

*442: Correct to: may also weaken*
**Reply**: done.

*718: correct to: Rb-Sr*
**Reply**: done.

*Fig. 1b: Where is the Naif thrust in the map? Furthermore, "calcschist with ophiolite": misfit of color between legend and map.*
**Reply**: we added the Naif thrust in the map and corrected all the misfit colour errors.

*Fig. 2: In lower hemisphere diagrams: increase size of Lm and poles to Sm.*
**Reply**: we increased the size of Lm and poles to Sm in lower hemisphere diagrams.

*Fig. 3: If possible, add approximate field orientations to all field photographs. Fig. 3h: Enlarge inset, it is too small to recognize calcite. Fig. 3d: The amphibolitic boudin does not look greenish (or arrow head is not exactly located?).*
**Reply**: done; we highlighted the amphibolitic boudin with dashed lines.

*Fig. 4: Write sample nos. on these photomicrographs.*
**Reply**: done.

*Figs. 7 - 9: It remains unclear which ages are actually listed in the Ar release patterns/age spectra on the left side. What means the stippled circles in the diagrams on the right side (T vs. Differential Release). Is there any information, from which mineral phase the Ca in all these patterns is coming from? Did you try calculate plateau and/or isotope inversion ages? Explain abbreviation EMPA in section 4.1.*
**Reply**: The alteration phases may give both younger and older apparent step ages; what is important is the observation that alteration phases have a non-stoichiometric Ca/Cl/K signature. The stippled circles highlight the peaks of the differential release (cfr. Villa, 2021) of Bt and Wm. We also explain the abbreviation of EMPA at line 333.

*Fig. 7: You could save much space writing the sample nos. insude of diagrams. A minor point: Use same scale in the age vs. Ca/K diagrams. Add information, which sort of $^{40}Ar/^{39}Ar$ ages are listed on diagrams and was the younger age of the first steps mean. Explain abbreviation EMPA in section 4.1.*
**Reply**: done.

---

## Referee Report (RR1)

**General remarks:**

The manuscript is dealing with the eo-Alpine ductile Vinschgau shear zone, a thrust, and proposed a well-developed state-of-the art combination of structural work and Ar-Ar geochronology associated with with detailed and extensive electron microprobe work. The work could represent a model for future work on ductile shear zones.

During revision, the authors followed all comments and suggestions of the reviewer and explained all details, so that the reader can easily follow their arguments.

I recommend publication as it is, and the few typos etc. could be solved during proof-reading.

**Specific remarks**

Figure 1A: I suggest to use in legend of A) to use singular: „Permian-Jurassic cover"

L. 67: use plural for „zone": „Large scale thrust- or normal-sense shear zones …."

L. 71: „features from shallow depths conditions": „depth": use singular

L. 149-150: Make clear that the age of 450 Ma is for the Partschinser orthogneiss.

L. 232-233: „VSZ studied transects": better: studied VSZ transects

L. 239: „basing": better: „based"

L. 277: „Correct to „Phyllosilicate domains"

L. 278: „The abundance of calcite increases" (not increase)

L. 347: You mean here Fig. 6a? (Fig. Xa)

L. 518-519: „Meliata-Hallstatt Ocean": Potentially omit Hallstatt. The Hallstatt facies is now generally seen as outer shelf facies/upper continental slope facies.

L. 525: „at least 7-8 Ma before the pressure peak recorded by the Texel eclogites": This open the question, whether the shear zone operated, at the Schlanders section, as thrust shear zone or as ductile normal normal fault at 97 Ma, when the eclogite-bearing Texel was down-doing to mantle depths. Potentially, these distinct processes are superposed on each other. However, this is a task for future work when more data are created for the Texel unit allowing a more detailed burial and exhumation history of the Texel unit.

L. 616: Correct to Franz (not Frantz)

L. 822: Sample (upper case).

---

## Author Response (AR2)

Dipartimento di Scienze dell'Ambiente e della Terra

*Piazza della Scienza 4*

*I-20126 Milano*

**Manuscript no. egusphere-2023-126**

Dear Editor,

please find attached the R2 revised version of our manuscript:

"**Kinematics and time-resolved evolution of the main thrust-sense shear zone in the Eo-Alpine orogenic wedge (the Vinschgau Shear Zone, Eastern Alps)**"

authored by

*Chiara Montemagni, Stefano Zanchetta, Martina Rocca, Igor M. Villa, Corrado Morelli, Volkmar Mair and Andrea Zanchi*

We wish to thank the reviewer Franz Neubauer and the topic editor Yang Chu for their careful suggestions and corrections.

In the following notes we account and answer to all the reviewer and editor comments, which have been accepted and incorporated in the text. All changes made to the text are highlighted in the "tracked changes" version of the manuscript together with the R1 changes.

We also modified the figure 1 in order to satisfy reviewer and editor requests.

We hope that the revised manuscript will be judged improved and will be suitable for publication in Egusphere: Solid Earth.

Sincerely yours

Chiara Montemagni, on behalf of all co-authors

Milano, 26 April 2023

**Changes to figures**

**Figure 1** has been modified according to reviewer and editor suggestions. In Fig. 1a in the legend we amended "Permian-Jurassic covers" with "Permian-Jurassic cover"; in Fig. 1c we moved the arrow pointing at the VSZ to avoid misunderstanding.

**Figure 2** has been replaced as it was a typo in the previous version: we corrected "Vermoi Spitze(2923 m)" to "Vermoi Spitze (2923 m)".

**Reviewer Franz Neubauer**

*The manuscript is dealing with the eo-Alpine ductile Vinschgau shear zone, a thrust, and proposed a well-developed state-of-the art combination of structural work and Ar-Ar geochronology associated with detailed and extensive electron microprobe work. The work could represent a model for future work on ductile shear zones.*
*During revision, the authors followed all comments and suggestions of the reviewer and explained all details, so that the reader can easily follow their arguments.*
*I recommend publication as it is, and the few typos etc. could be solved during proof-reading.*

*Reviewer: L. 67: use plural for „zone": „Large scale thrust- or normal-sense shear zones ...."*
**Reply:** done

**Reviewer:** L. 71: „features from shallow depths conditions": „depth": use singular
**Reply:** done

*Reviewer: L. 149-150: Make clear that the age of 450 Ma is for the Partschinser orthogneiss.*
**Reply:** done

*Reviewer: L. 232-233: „VSZ studied transects": better: studied VSZ transects*
**Reply:** done

*Reviewer: L. 239: „basing": better: „based"*
**Reply:** done

*Reviewer: L. 277: „Correct to „Phyllosilicate domains"*
**Reply:** done

*Reviewer: L. 278: „The abundance of calcite increases" (not increase)*
**Reply:** done

*Reviewer: L. 347: You mean here Fig. 6a? (Fig. Xa)*
**Reply:** Yes, we amended it.

*Reviewer: L. 518-519: „Meliata-Hallstatt Ocean": Potentially omit Hallstatt. The Hallstatt facies is now generally seen as outer shelf facies/upper continental slope facies.*
**Reply:** done

*Reviewer: L. 525: „at least 7-8 Ma before the pressure peak recorded by the Texel eclogites": This open the question, whether the shear zone operated, at the Schlanders section, as thrust shear zone or as ductile normal normal fault at 97 Ma, when the eclogite-bearing Texel was down-doing to mantle depths. Potentially, these distinct processes are superposed on each other. However, this is a task for*

*future work when more data are created for the Texel unit allowing a more detailed burial and exhumation history of the Texel unit.*

**Reply:** we agree with the reviewer that the current data set does not allow to recognize if the first stage of activation of the VSZ around 95 Ma occurred in an extensional or contractional deformation regime. More geochronological and structural data about the P-T-t evolution of the Texel and Schneeberg units are mandatory to disentangle this problem.

*Reviewer: L. 616: Correct to Franz (not Frantz)*
**Reply:** done, we are sorry for this mistake.

*Reviewer: L. 822: Sample (upper case).*
**Reply:** done

**Topic editor Yang Chu**

*This manuscript has been carefully revised and is almost ready for publication. As suggested by one reviewer, there are still some flaws to be fixed. I also give some suggestions below for minor revisions.*

*Editor: Line 19: Replace age by timing.*
**Reply:** done

*Editor: Line 20: Dip is not clear here. Dip angle can increase, but I do not know how dip increases from W to E.*
**Reply:** done

*Editor: Line 23: Top to the W equals to W-directed.*
**Reply:** done

*Editor:Line 29: Delete direction.*
**Reply:** done

*Editor: Figure 1c: The valley pointed by VSZ is Schnals, nor Vinschgau.*
**Reply:** done

*Editor: Line 79: I do not know why discussing this topic, unless there are some sentences on how large shear zones develop in collisional belts.*
**Reply:** we modified "large" with "regional" as "large" could generate misunderstanding: the VSZ is one of the main regional scale shear zone in the Alps for its continuity of exposure both along strike and across the entire shear zone.

*Editor: Line 445: This sentence is ambiguous. The largest shear zone in Alps is only tens of kilometers long?*
**Reply:** At lines 57-62 we have reported other important shear zones in the Alps: the VSZ in one of the largest and important shear zone in the Alps as it exhumed Eo-Alpine Austroalpine wedge over the Pre-Alpine tectonometamorphic units, so it is a major tectonic and metamorphic boundary. Anyway we modified a little the sentence.